# Estimating the mortality burden of large scale mining projects—Evidence from a prospective mortality surveillance study in Tanzania

**Isaac Lyatuu**[1,2,3]*, **Mirko S. Winkler**[2,3], **Georg Loss**[2,3], **Andrea Farnham**[2,3], **Dominik Dietler**[2], **Günther Fink**[2,3]

**1** Ifakara Health Institute, Dar es Salaam, United Republic of Tanzania, **2** Department of Epidemiology and Public Health, Swiss Tropical and Public Health Institute, Basel, Switzerland, **3** University of Basel, Basel, Switzerland

* ilyatuu@ihi.or.tz

**Data Availability Statement:** The population level data (census data) and global burden of disease data are publicly available from Tanzania national

## Abstract

We set up a mortality surveillance system around two of the largest gold mines in Tanzania between February 2019 and February 2020 to estimate the mortality impact of gold mines. Death circumstances were collected using a standardized verbal autopsy tool, and causes of death were assigned using the InSilicoVA algorithm. We compared cause-specific mortality fractions in mining communities with other subnational data as well as national estimates. Within mining communities, we estimated mortality risks of mining workers relative to other not working at mines. At the population level, mining communities had higher road-traffic injuries (RTI) (risk difference (RD): 3.1%, Confidence Interval (CI): 0.4%, 5.9%) and non-HIV infectious disease mortality (RD: 5.6%, CI: 0.8%, 10.3%), but lower burden of HIV mortality (RD: -5.9%, CI: -10.2%, -1.6%). Relative to non-miners living in the same communities, mining workers had over twice the mortality risk (relative risk (RR): 2.09, CI: 1.57, 2.79), with particularly large increases for death due to RTIs (RR: 14.26, CI: 4.95, 41.10) and other injuries (RR:10.10, CI: 3.40, 30.02). Our results shows that gold mines continue to be associated with a large mortality burden despite major efforts to ensure the safety in mining communities. Given that most of the additional mortality risk appears to be related to injuries programs targeting these specific risks seem most desirable.

## Introduction

Extractive industry projects have the potential to trigger improvements in socio-economic status and public infrastructure at the local level, along with generating tax and royalty incomes at the national and sub-national levels [1]. This potential source of development is particularly relevant for the African continent, which is endowed with over 30% of the world's global mineral reserves [2] and features the highest rates of poverty globally [3].

Given the importance of large mining operations for regional economic development, mining also has the potential to improve other indicators targeted by the Sustainable Development Goals (SDGs)–most importantly population health. Recent studies have found positive links

bureau of statics website (https://www.nbs.go.tz) and global burden of disease website (https://vizhub.healthdata.org/gbd-compare/) respectively. Mortality data is available upon request from the Tanzania Ministry of Health Community Development, Gender, Elderly and Children (MoHCDGEC). The mortality data is not publicly available because it contains individual level information. Making this data publicly available would breach compliance with the protocol approved by the local research ethics board.

**Funding:** This research received funding from the Swiss Programme for Research on Global Issues for Development (r4d Programme, www.r4d.ch (Access On 10th February 2021)), which is a joint funding initiative of the Swiss Agency for Development and Cooperation (SDC) and the Swiss National Science Foundation (SNSF) (grant number 169461, grant recipient: MW). The funders had no role in the study design, data collection and analysis, decision to publish, or preparation of the manuscript.

**Competing interests:** The authors declare that they have no conflict of interest.

between resource rents and life expectancy at the country level [4], and linked resource extraction projects to reduced prevalence of undernutrition and infectious disease [5, 6] as well as to reduced incidence of acute and chronic health conditions [7]. Evidence from Ghana [8, 9], Tanzania [9] and Sub-Saharan Africa (SSA) at large [10] also suggests lower infant mortality in regions with mining projects.

On the other hand, mining has long been identified as a hazardous industry with often substantially increased risk of adverse health outcomes for miners and surrounding communities [11]. Studies have linked mining to increased levels of cancer [12–16], poisoning [17], cardiovascular diseases [18, 19], respiratory diseases [19–22] and adverse pregnancy outcomes [23, 24], as well as injuries [25, 26] and tuberculosis (TB) [21, 27].

Chronic exposures to toxic substances, poor air quality and noise pollution have been highlighted as key mechanisms underlying these adverse health effects. This also applies to gold mining, independent of whether the metal is extracted in industrial mines [28–31] or through artisanal and small-scale gold mining [32, 33]. Given the large health risks documented historically, major efforts have been made in recent years to establish environmental and occupational safety protocols including the establishment of the Occupational Safety and Health Authority (OSHA) in 2001 and the formulation of National Occupational Health and Safety Policy in 2010 [34]. These efforts are generally supported by the large international corporations that run most industrial mines. In this paper, we focus on gold mining, which is the largest mining sector in Tanzania, accounting for 88% of all mineral export and 8% of national income in 2019 [35]. The presence of large-scale international mining companies in Tanzania presents an opportunity to improve the livelihoods of local communities while ensuring the safety of their workers, but it remains unclear to what degree international safety measures and national level efforts have actually made gold mining safer.

To be able to assess the current safety of gold mining activities in Tanzania and to reconcile the conflicting current evidence on the health impact of mines, we established active mortality surveillance systems based on verbal autopsy in two of the largest gold mining areas in Tanzania in 2019 and closely monitored mortality outcomes over a 12-month period.

## Methods

### Study design

This is a prospective population cohort study designed to monitor mortality outcomes over 12 months in purposively selected communities surrounding two major gold mines in Tanzania.

### Study areas

The study was conducted in two mining areas: (1) the Geita Gold Mine (GGM) in Geita Town Council (TC), Geita Region; and (2) the Bulyanhulu Gold Mine (BGM) in Msalala District Council (DC), Shinyanga Region.

GGM and BGM are both located in the Tanzanian mainland along the Lake Victoria gold belt (Fig 1). The Victoria gold belt is located in the northern part of Tanzania and is where most gold extraction takes place in Tanzania. This area hosts Tanzania's major multi-national gold extraction companies, as well as artisanal and small-scale mining activities. Besides mining, the primary economic activities of these areas are crop and livestock agriculture [36].

GGM and BGM are approximately 70 km apart. GGM was opened in 2000 and is located in Geita Town, which constitutes one of six administrative Districts in the Geita Region. BGM was opened in 2001 and is located in Msalala District, which is also one of six districts in the Shinyanga Region. Geita Town and Msalala Districts' estimated populations were 192,707 and 250,727 people, in 2019 respectively [37], which is equivalent to 11% and 16% of the respective

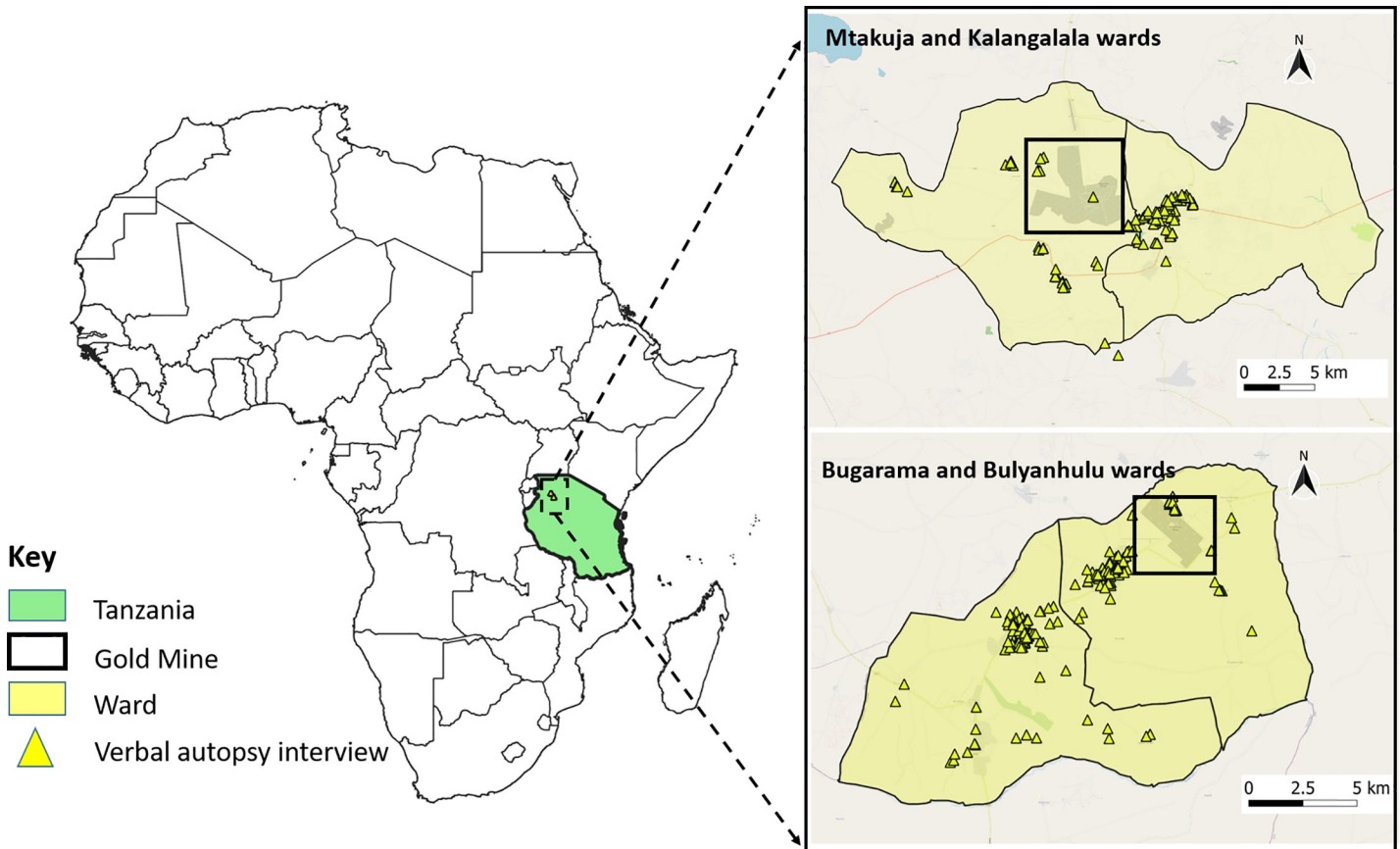

**Fig 1. Location of the mining areas and the study sites (left) including location where the verbal autopsy interview occurred (right).** This figure was created using QGIS, an open-source application. The source basemap was obtained using OpenStreetMap plugin in QGIS. OpenStreeMap follows open data license under Open data Commons Open Database Licence (ODbL, https://openstreetmap.org/copyright).

regional population. GGM is an open-pit mine and BGM is an underground mine. At the time of data collection, GGM was operating at full capacity, while BGM was operating at reduced capacity (about 10% production, [38]). In addition to the industrial mining, artisanal activities are common in both study areas.

We used satellite images to identify and select community settlements around the two gold mines. Tanzania is administratively divided into 30 regions and 169 districts or councils, which are further divided into wards [39]. Wards have an average population of around 13,000 people [39]. Our study communities fall within the following administrative wards: (1) Kalangalala, (2) Mtakuja and (3) Mgusu (formally part of Mtakuja) wards in Geita TC; and (4) Bugarama and (5) Bulyanhulu wards in Msalala DC. Kalangalala, Mtakuja, Mgusu and Bulyanhulu Wards are all located next to the industrial mining center. Bugarama Ward is located west of the main mining ward. The ward was chosen because it contains the main access road to Bulyanhulu mine, as well as having a community settlement for miners. The east side of Bulyanhulu gold mine is mostly unoccupied with a limited number of households. Population estimates of the wards are provided in Table 1.s

## Mortality data

To monitor mortality in the selected wards, we employed one community health care worker (CHW) for each of the selected wards. Each CHW was given a tablet installed with the Open

Table 1.  Population under surveillance around Geita gold mine and Bulyanhulu gold mine.

| Location | Total Population 2019 | Population (15–64 Age Group) | | Working at Mines | | Not Working at Mines | |
|---|---|---|---|---|---|---|---|
| | | male | female | male | female | male | female |
| **Geita Gold Mine** | | | | | | | |
| Mgusu ward | 16,221 | 4,148 | 4,493 | 1,606 | 953 | 2,542 | 3,540 |
| Kalangalala ward | 16,660 | 4,260 | 4,615 | 1,649 | 979 | 2,611 | 3,636 |
| Mtakuja ward | 52,812 | 13,505 | 14,629 | 5,228 | 3,103 | 8,277 | 11,526 |
| **Bulyanhulu Gold Mine** | | | | | | | |
| Bulyanhulu ward | 27,854 | 7,123 | 7,716 | 2,757 | 1,637 | 4,365 | 6,079 |
| Bugarama ward | 20,092 | 5,138 | 5,566 | 1,989 | 1,181 | 3,149 | 4,385 |
| | **133,639** | **34,173** | **37,018** | **13,228** | **7,852** | **20,945** | **29,166** |

Data Kit (ODK) and verbal autopsy (VA) tool, and trained on the overall objectives of the study as well as on how to conduct verbal autopsy interviews using ODK. VA implementation involves interviewing a close relative of the deceased using a structured questionnaire in order to capture sequence of events and symptoms leading to death [40–42]. The establishment of the possible cause of death is then derived using physician review or computational approaches [43–45]. For our study, we used the standard VA instrument of the World Health Organization [46] version 1.5.1 adapted to Tanzania administrative structures [47]. The tool has been translated to Swahili (local language) and also applies local Tanzania administrative structures. The tool is designed for all age groups, including maternal and neonatal deaths, as well as deaths caused by injuries.

We conducted community sensitization to describe the project and explain the importance and significance of reporting death events. During the community sensitizations, we recruited community members as death notification officers. The sensitization was followed by active monitoring and notification of death events by community members to CHW in each of the selected wards. Both community and facility deaths were notified. CHW followed and conducted VA interviews on notified death events two weeks after event notification. CHW sought formal verbal consent to each VA interview. Consent information was captured and stored on tablet along with other variables. The VA interview lasted between 45 to 60 minutes. The interview data along with consent information were transmitted directly from the tablet to the Ministry of Health central server under the Civil Registration and Vital Statistics (CRVS) unit. We provided summary information of collected interviews to CHW and provided supportive supervision at the council level [48]. The VA interviews were conducted on deaths that occurred in a period of one year, from February 2019 to February 2020.

**VA data processing.**   We used InSilicoVA version 1.3.0 [45] to obtain the underlying cause of death information for each of the VA questionnaire completed. InSilicoVA employs probabilistic methods [49] to calculate cause-specific mortality fractions against 67 VA target causes (S1 Appendix) from VA interviews. We grouped cause of death information from InSilicoVA into seven broad mortality categories defined as (1) Infectious diseases other than HIV, (2) HIV/AIDS-related deaths, (3) Road Traffic Injuries (RTIs), (4) Injuries other than RTIs, (5) Cardiovascular diseases, (6) Cancer/neoplasm and (7) Other.

## Variables

The VA tool contains variables on signs, symptoms or conditions that led to the death of an individual. For our study-specific needs, we added a binary indicator to capture whether the deceased individual ever worked in the mining sector prior to death. We captured deaths that occurred at home as well as in health facilities.

## Population data

We obtained population data for 2019 from the Tanzania 2012 census projections [37]. We calculated the number of people who work in the mining sector using a percentage of the exposed population from the Maliganya & Paul [36] study. We estimated the total number of people who work in the mining sector by multiplying the percentage of people who work in the mining sector in Geita Region [36] and the 2019 ward population using national census data projections [37].

## Other data

We used two additional data sources to compare the overall distribution of causes of deaths: (1) mortality estimates from the Iringa VA demonstration site [50] in Iringa Region, Tanzania, which contains all reported deaths in the region in 2019 and (2) Tanzania's national 2019 mortality estimates from the Global Burden of Disease (GBD) project (https://vizhub.healthdata.org/gbd-compare/). The Iringa Region is located in the southern highlands of Tanzania and is a part of the Tanzania CRVS technical feasibility study for large scale roll-out of the VA methods and CRVS systems integration. The Iringa Region is currently the only region in Tanzania that has a full coverage of VA implementations and thus a natural benchmark for the VA data collected and analysed in this project. Iringa region has a total of 3,238,347 population, 106 Wards and five administrative districts [37]. At the national level, the only available estimates are from GBD.

## Statistical analysis

We limited our analysis to deaths that occurred in 2019. We calculated cause-specific mortality fractions based on the seven broad mortality categories defined for this project and available in all data sets. The seven broad mortality categories are 1) HIV/AIDS related deaths, 2) Road Traffic Injuries (RTIs), 3) Injuries other than RTIs, 4) Cardiovascular diseases, 5) Infectious diseases other than HIV, 6) Cancer and 7) Other. In the first step, we summarized the causes of deaths in the VA data by age group and sex and compared the distribution of causes of death in the surveyed areas to national estimates from the GBD and regional estimates from the Iringa pilot.

In a second step, we focused on deaths occurred around the two mines, and compared individuals that directly worked in mines to those in the same communities but were not actively engaged in mining. We estimate overall relative mortality risk as well as relative mortality risk for each category.

For this second analysis, we restricted mortality to the working ages 15–64.

## Ethical considerations

This study obtained ethical approval from Ifakara Health Institute Review Board and the National Institute for Medical Research (NIMR) in Tanzania, the Ethics Committee of Northwestern and Central Switzerland (Ethikkommission Nordwest- und Zentralschweiz, EKNZ) and the institutional review board of the Swiss Tropical and Public Health Institute (Swiss TPH) in Switzerland.

**Consent to participate.** Interviews were conducted after obtaining formal verbal consent from the study participant.

## Results

### Population estimates

Table 1 summarizes the estimated age and gender composition of populations living in the five wards under surveillance. The total estimated population in the five wards was 133,639

**Table 2. Number of deaths by age group, sex and cause of death.**

| Cause of Death | Under 5 | | Children (5–14) | | Adults (15–64) | | Senior (65+) | | Grand Total |
|---|---|---|---|---|---|---|---|---|---|
| | Male | Female | Male | Female | Male | Female | Male | Female | |
| Infectious diseases other than HIV | 19 | 12 | 5 | 5 | 25 | 7 | 8 | 8 | 89 |
| Cardiovascular diseases | 3 | 1 | 1 | 1 | 20 | 7 | 10 | 18 | 61 |
| HIV/AIDS related death | 0 | 0 | 0 | 0 | 17 | 17 | 4 | 1 | 39 |
| Road Traffic Injuries | 0 | 0 | 0 | 0 | 26 | 2 | 0 | 1 | 29 |
| Cancer | 0 | 0 | 0 | 0 | 5 | 12 | 8 | 3 | 28 |
| Injuries other than RTI | 3 | 0 | 0 | 1 | 17 | 4 | 1 | 0 | 26 |
| Other | 14 | 14 | 0 | 7 | 13 | 14 | 10 | 5 | 77 |
| All Deaths | 39 | 27 | 6 | 14 | 123 | 63 | 41 | 36 | 349 |

people. Of the total working-age population, 30% (N = 21,080) were estimated to work in the mining sector [36]. 37% of mining workers were female (N = 7,852).

Table 2 summarizes the total number of deaths. A total of 349 deaths occurred in the study areas in the year 2019 (209 male, 160 female). Sixty-six deceased were under the age of five (Under 5), 20 were between the age of 5 and 14 (children), 186 were of working age (15 to 64), and 77 were 65 years and above. Of the 186 working age deaths, 87 (47%) worked in the mining sector. The mean age of the deceased who worked in the mine was found to be 43 years, with a minimum of 21 years and a maximum of 84 years. Deceased seniors who worked in the mine were 3 females and 10 males.

Table 2 also summarizes the main causes of death. Twenty-six percent (N = 89) of deaths were due to infectious diseases other than HIV, 17% (N = 61) were due to cardiovascular diseases, 11% (N = 39) were due to HIV/AIDS-related deaths, 8% (N = 29) were due to RTIs, 8%, (N = 28) were due to cancer and 7% (N = 26) were due to injuries other than RTIs.

Fig 2 shows cause-specific mortality fractions (CSMF) for mining areas, as well as the GBD estimates and Iringa estimates. In mining sites, the three most common causes of death were infectious disease other than HIV, cardiovascular disease, and HIV/AIDS related deaths. The Iringa ranking was very similar overall, with a switch in the first two categories. Relative to these estimates, the main difference in the GBD ranking was the third place for cancer, which only appeared as the 4th and 5th most common category in the mining and Iringa data.

Fig 3 shows more CSMF differences between mining areas and the Iringa region. Relative to the Iringa region, the share of death due to non-HIV infectious diseases and RTIs was 5.56% (95% confidence interval (CI): 0.01, 0.10) and 3.14% (CIs: 0.004, 0.059) higher in mining sites. The share of HIV/AIDS related deaths was 5.93% (CIs: - 0.102, - 0.016) lower in mining sites compared to the Iringa region.

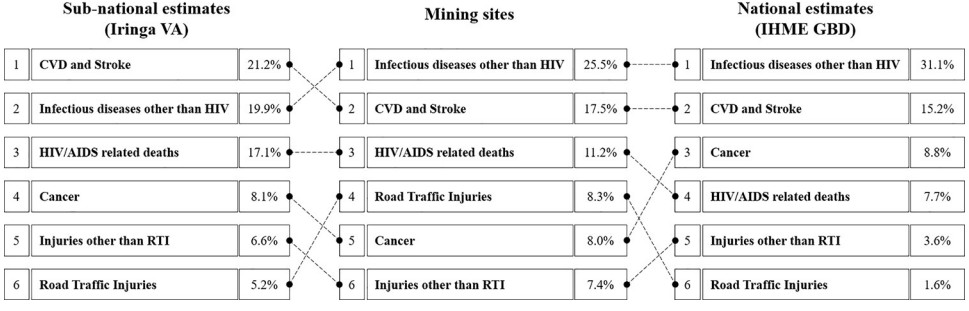

**Fig 2. Causes of death comparison.**

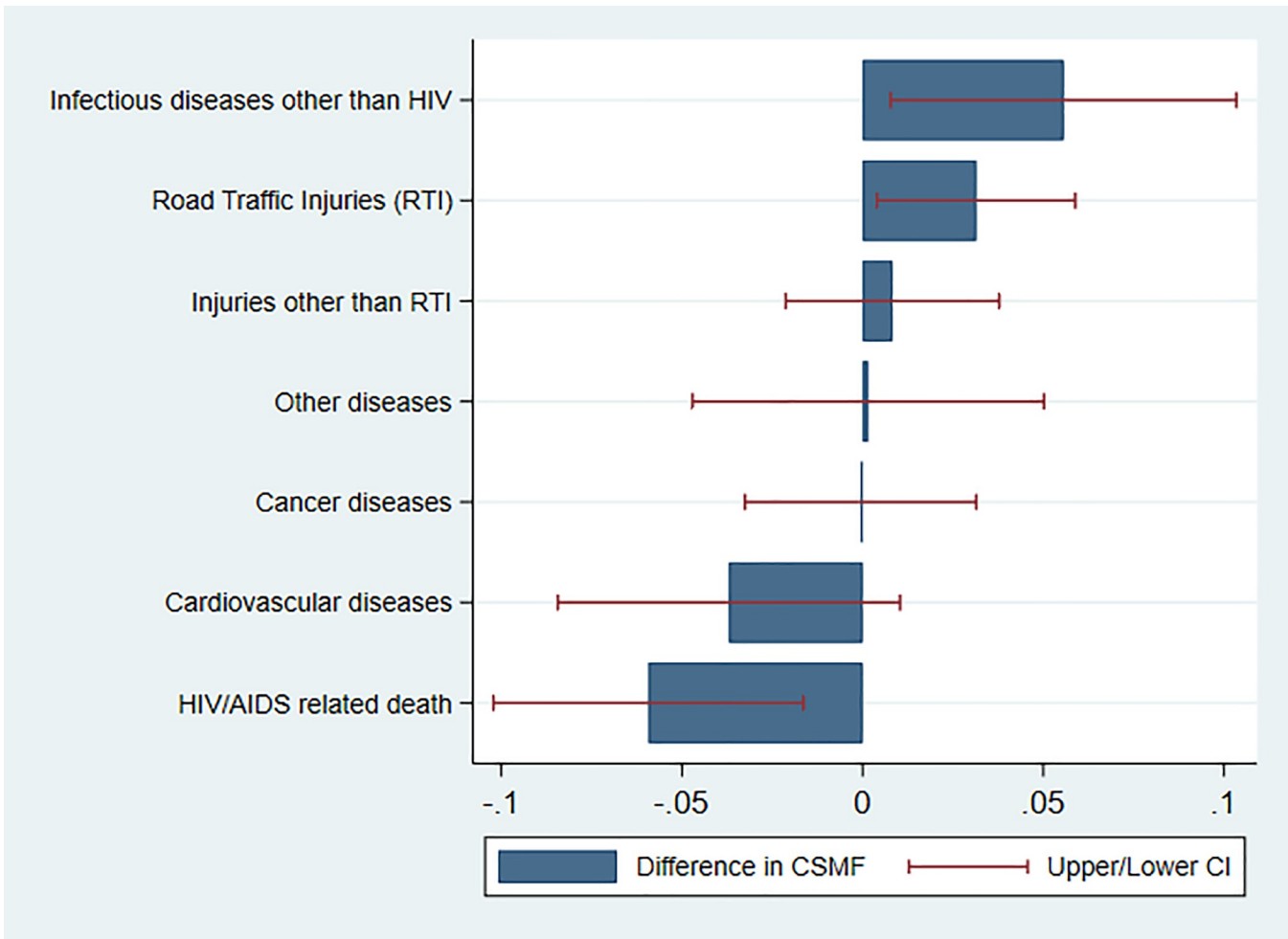

**Fig 3. Differences in CSMF between mining sites and Iringa region.**

Table 3 shows the number of working age deaths in each category separately for individuals working and not working at mines. Eighty-seven deaths were reported among mining workers, 89% (N = 77) were male, 11% (N = 10) were female.

Table 4 shows relative risk (RR) estimates for mining workers living in the mining area. Compared to working age residents not working at mines, mining workers had twice the overall risk of death (RR = 2.09, CI: 1.57, 2.79), with particularly large increases for RTIs and injuries unrelated to RTIs for males. Men working at mines had 2.65 times higher risk of dying compared to non-exposed men (RR = 2.65, CI: 1.84, 3.82), while no such associations were found for women (RR: 0.70, CI: 0.36, 1.38). Men working at mines had almost three times the risk of dying from HIV/AIDS-related causes (RR = 2.90, CI: 1.07, 7.85), 14 times higher risk of dying from RTIs (RR = 14.26, CI: 4.95, 41.10) and 10 times higher risk of dying from other injuries (RR = 10.10, CI: 3.40, 30.02).

## Discussion

In this study, we prospectively monitored mortality in communities neighboring two major gold mines in Tanzania for a full year to assess the overall impact of mining operations on

**Table 3. Mortality by mining affiliation.**

| | Working at mines | | Not Working at Mines | | Total |
|---|---|---|---|---|---|
| | **Male** | **Female** | **Male** | **Female** | |
| **Population** | **13,228** | **7,852** | **20,945** | **29,166** | **71,191** |
| **Cause of Death Information** | | | | | |
| HIV/AIDS–related | 11 | 3 | 6 | 14 | 34 |
| Road Traffic Injuries | 24 | 0 | 2 | 2 | 28 |
| Injuries other than RTI | 16 | 1 | 1 | 3 | 21 |
| Non-communicable disease | 10 | 1 | 10 | 6 | 27 |
| Inf. diseases other than HIV | 8 | 0 | 17 | 7 | 32 |
| Cancer | 4 | 2 | 1 | 10 | 17 |
| Other | 4 | 3 | 9 | 11 | 27 |
| **Total** | 77 | 10 | 46 | 53 | 186 |

population health. While the results from the VA data analysis show that overall mortality patterns in the mining communities were relatively similar to the subnational estimates from the Iringa VA implementation study and national estimates using GBD numbers, we found that mining areas had a higher incidence of mortality from RTIs and non-HIV infectious diseases and a lower incidence of HIV mortality compared to other parts of Tanzania, as well as Tanzania overall. In terms of absolute risk, we found that overall mortality was significantly elevated among miners, particularly among male mine workers, who had more than 10 times the mortality risk due to RTIs and other injuries unrelated to RTIs.

Mortality from road traffic injuries is increasingly recognized as central threat to population health in low- and middle-income countries (LMIC). The guide for road safety opportunities and challenges report estimates 93% of RTIs occur in LMIC [51]. In Tanzania, the majority of RTIs appear to be due to motorcycle accidents commonly known as bodaboda [52, 53]. The presence of mining activities can contribute to rapid population growth and urbanization through the boomtown effect [54], as well as by increasing access and affordability of motorcycles and other vehicles due to improvements in socio-economic status in mining areas [55]. Hence, these two factors might partly explain the higher mortalities in relation to RTIs in the mining areas compared to non-mining areas, as well as the tenfold increase in the risk of dying from RTIs in people who worked in gold mining. Men in particular showed a much higher risk of RTI-related death. This may be due to the nature of transport-related activities, which is a male dominated occupation. Our results also show that the risk of dying from other injuries

**Table 4. Relative risk of cause-specific mortality among mining workers living in the surveillance area.**

| Outcome: | All | | Male | | Female | |
|---|---|---|---|---|---|---|
| | **RR** | **95% CI** | **RR** | **95% CI** | **RR** | **95% CI-L** |
| **All-cause mortality** | **2.09** | **(1.57,2.79)** | **2.65** | **(1.84,3.82)** | 0.70 | (0.36,1.38) |
| HIV/AIDS-related deaths | 1.66 | (0.84,3.29) | **2.90** | **(1.07,7.85)** | 0.80 | (0.23,2.77) |
| Road Traffic Injuries | **14.26** | **(4.95,41.10)** | **19.00** | **(4.49,80.38)** | * | |
| Injuries other than RTI | **10.10** | **(3.40,30.02)** | **25.33** | **(3.36,191.01)** | 1.24 | (0.13,11.9) |
| Cardiovascular diseases | 1.63 | (0.76,3.52) | 1.58 | (0.66,3.8) | 0.62 | (0.07,5.14) |
| Inf. diseases other than HIV | 0.79 | (0.36,1.76) | 0.75 | (0.32,1.73) | * | |
| Cancer | 1.30 | (0.48,3.51) | 6.33 | (0.71,56.66) | 0.74 | (0.16,3.39) |
| Other | 0.83 | (0.35,1.97) | 0.70 | (0.22,2.28) | 1.01 | (0.28,3.63) |

unrelated to RTIs is significantly higher among former miners in the mining communities, particularly in men. This finding is similar to previous literature linking mining activities and increased injury outcomes [56–59]. It is likely that interventions such as community road safety programs that incorporate local community perspectives can contribute to reductions in mortality in mining areas [60].

Our findings contrast with data from previous studies that have documented a significantly higher risk of HIV/AIDS in mining communities [61, 62]. HIV transmission is commonly hypothesized to be elevated in mining areas due to the access to higher income and distance from home among the miners, increasing the likelihood of high risk sexual behaviors [61]. The reason why our findings show comparatively low numbers of HIV/AIDS-related deaths in the mining communities could be due to the contribution of health and education programs often associated with the presence of gold mining industries [5, 63, 64]. We did however find that males who lived in the mining communities and worked in the gold mining sites have about three times the risk of dying of HIV compared to men of similar age not working at mines. The increased overall mortality risk among men working in the mines is not surprising when considering that the mining industry is a male-dominated sector, with women pursuing less dangerous activities [65].

Overall, the leading causes of death observed in the two gold mining communities are (1) infectious diseases other than HIV (including acute respiratory infections, tuberculosis, asthma and chronic obstructive pulmonary diseases); (2) Cardiovascular diseases; and (3) HIV/AIDS related deaths. These shares account for more than half of all deaths in the study population. Respiratory infections as a leading cause of death seem quite plausible overall given that air pollution in gold mining has been linked to the spread of pathogens and potentially toxic elements which can contribute to the increase of respiratory diseases [66, 67]. Our findings are similar to previous literature that found an association between gold mining and increased respiratory diseases [15, 18, 19] and asthma [68]. In addition, a recent qualitative study involving focus group discussions from communities around GGM and BGL published similar concerns with regards to air pollution coming from large-scale mining activities [69]. Given that mining is strongly linked to environmental pollution [18, 22], additional steps such as the application of health impact assessment prior to the development of new mining projects [70, 71] and the incorporation of infectious disease risk assessment and management plans [72] into corporate social responsibility can help to minimize the observed pollution-related disease burden.

## Strength and limitations

The main strength and novelty of this study is the prospective establishment of a mortality surveillance system in the mining community to monitor mortality in a low income setting where vital registration systems are largely lacking. We managed to build capacities of local CHWs to collect and manage VA data. Despite these efforts, a substantial number of deaths may not have been reported. While the true number of deaths is not known, we estimate that about 50–60% of deaths were reported by applying the current national crude death rates to our population. Factors that influence incomplete reporting include the relatively large size of wards each agent was responsible for and the remoteness of some areas. To minimize these data completeness issues, we conducted supervision of fieldwork activities and encouraged routine sharing of the fieldwork progress to the local administration office. Overall, we believe the data presented here represent the most comprehensive cause-specific mortality estimates available to date.

In addition, it was not possible to distinguish population level exposures of the artisanal and industrial mining. Our study was not large enough to look at specific diseases.

Furthermore, it was limited to a smaller follow-up period. The seven groups chosen reflect major disease groups–much larger studies would be needed to look at the incidence of specific diseases such as various types of cancers or poisoning that could be related to mining. Finally, our study was limited by the lack of a directly comparable population. While the data from Iringa are comparable in terms of the data collection tool, socio-economic, climatic and geographic factors are likely different around both mines and thus limit the comparability of the population level estimates.

## Conclusion

The results presented in this paper suggest that mining areas are relatively similar to other areas of Tanzania in terms of the primary causes of death, and do not show higher rates of HIV mortality as suggested by some of the literature. Relative to other adult community members, mining workers–and male mining workers in particular—appeared to have substantially increased mortality risk. Most of this increased mortality risk appears to be due to RTIs and other injuries unrelated to RTIs. Programs targeting these areas may be needed to further improve the overall health impact of major mining operations.

## Supporting information

**S1 Appendix.**
(DOCX)

## Acknowledgments

The authors would also like to thank Prof. Dr. Don de Savigny for useful comments on early drafts of this work. The authors would also like to thank Mr. Gisbert Msigwa, the national coordinator of the Civil Registration and Vital Statistics program at the Ministry of Health, in Tanzania for supporting project activities throughout. Last but not least, the authors would like to thank the Tanzanian Ministry of Health, Community Development, Gender, Elderly and Children for allowing the use of VA data from the Iringa VA demonstration phase.

## Author Contributions

**Conceptualization:** Isaac Lyatuu, Günther Fink.

**Formal analysis:** Isaac Lyatuu.

**Funding acquisition:** Mirko S. Winkler.

**Methodology:** Isaac Lyatuu, Mirko S. Winkler.

**Supervision:** Mirko S. Winkler, Günther Fink.

**Writing – original draft:** Isaac Lyatuu.

**Writing – review & editing:** Isaac Lyatuu, Mirko S. Winkler, Georg Loss, Andrea Farnham, Dominik Dietler, Günther Fink.

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
