## [Decision Letter · Decision Letter 0]

1 Jul 2021

 PGPH-D-21-00132 Estimating the mortality burden of large scale mining projects – Evidence from a prospective mortality surveillance study in Tanzania PLOS Global Public Health

Dear Dr. Lyatuu,

Thank you for submitting your manuscript to PLOS Global Public Health. After careful consideration, we feel that it has merit but does not fully meet PLOS Global Public Health’s publication criteria as it currently stands. Therefore, we invite you to submit a revised version of the manuscript that addresses the points raised during the review process.

We look forward to receiving your revised manuscript.

Kind regards,

Francesco Chirico, MD

Academic Editor

Journal Requirements:

Reviewers' comments:

Reviewer's Responses to Questions

**Comments to the Author**

1. Does this manuscript meet PLOS Global Public Health's publication criteria? Is the manuscript technically sound, and do the data support the conclusions? The manuscript must describe methodologically and ethically rigorous research with conclusions that are appropriately drawn based on the data presented.

Reviewer #1: Yes

2. Has the statistical analysis been performed appropriately and rigorously?

Reviewer #1: Yes

3. Have the authors made all data underlying the findings in their manuscript fully available (please refer to the Data Availability Statement at the start of the manuscript PDF file)?

Reviewer #1: Yes

4. Is the manuscript presented in an intelligible fashion and written in standard English?

Reviewer #1: Yes

5. Review Comments to the Author

Reviewer #1: Dear Authors,

I read with interest this manuscript and I have some questions to pose to Authors.

1) Could you please describe in more detail what VA is? I think that general readers could not be aware of caractheristics of this tool. I think it can be added to the method section.

2) I cannot find Appendix A1, and thus I'm asking if a table showing how the 67 causes of death were coded in the 7 cathegories pointed out in the manuscript has been included. If not, I think that it might be useful.

3) It is not clear to me in which way the exposure in mines has been assigned. I mean, when and how long a person had to be a mine worker to be included among exposed people? I think it should be clarified and added to methods.

4) I think that the short follow-up period should be highlighted as a limit of this paper.

6. PLOS authors have the option to publish the peer review history of their article (what does this mean?). If published, this will include your full peer review and any attached files.

**Do you want your identity to be public for this peer review?** For information about this choice, including consent withdrawal, please see our Privacy Policy.

Reviewer #1: No

---

## [Editor Report · Decision Letter 1]

31 Aug 2021

 PGPH-D-21-00132R1 Estimating the mortality burden of large scale mining projects – Evidence from a prospective mortality surveillance study in Tanzania PLOS Global Public Health

Dear Dr. Lyatuu,

Thank you for submitting your manuscript to PLOS Global Public Health. After careful consideration, we feel that it has merit but does not fully meet PLOS Global Public Health’s publication criteria as it currently stands. Therefore, we invite you to submit a revised version of the manuscript that addresses the points raised during the review process.

 The authors in their cover letter have declared to accept all reviewer's comments, but not all the changes

(in yellow as indicated) have been highlighted in the revised version of the paper. The authors should submit a revised version of the manuscript with the highlighted changes.

We look forward to receiving your revised manuscript.

Kind regards,

Francesco Chirico, MD

Academic Editor
---

## [Editor Report · Decision Letter 2]

15 Sep 2021

Estimating the mortality burden of large scale mining projects – Evidence from a prospective mortality surveillance study in Tanzania

PGPH-D-21-00132R2

Dear Dr. Lyatuu,

We're pleased to inform you that your manuscript has been judged scientifically suitable for publication and will be formally accepted for publication once it meets all outstanding technical requirements.

Within one week, you'll receive an e-mail detailing the required amendments. When these have been addressed, you'll receive a formal acceptance letter and your manuscript will be scheduled for publication.

An invoice for payment will follow shortly after the formal acceptance. To ensure an efficient process, please log into Editorial Manager at https://www.editorialmanager.com/pgph/ click the 'Update My Information' link at the top of the page, and double check that your user information is up-to-date. If you have any billing related questions, please contact our Author Billing department directly at authorbilling@plos.org.

Kind regards,

Francesco Chirico, MD

Academic Editor